# Players' strategy selection in co-governance and supervision of internet platforms' monopolistic behaviors: A study on new media participation

Yadong Song, Tongshui Xia*

School of Business, Shandong Normal University, Ji'nan, China

* 117011@sdnu.edu.cn

## Abstract

Incidents of monopolies among Internet platforms have seriously endangered the development of the market economy, public interest, and social fairness, making it a highly discussed topic of broad public concern. Preventing such incidents requires not only a comprehensive supervision system by governments, but also contributions from other relevant parties. The new media environment has provided a new platform to support such joint supervision from multiple parties. As such, this study constructed an evolutionary game model involving the government, Internet platforms, new media, and the public to explore the stable equilibrium point of players' strategy selections. The stability of the strategy combinations was tested using Lyapunov's first stability method, and MATLAB 2021b was used to conduct simulation analysis of the impact of each decision variable on players' strategy selection. The results showed that (1) new media participation in co-governance and public complaints/reports facilitated government supervision; (2) government's application of co-governance and supervision and public complaints/reports promotes compliance by Internet platforms; (3) new media plays a supplementary role when government supervision is lacking; the greater the impact of new media, the greater its supervisory effect on Internet platforms; and (4) effective reduction of costs stimulates the enthusiasm of the government and new media, and increases the success of the anti-monopoly co-governance and supervision system. Measures and suggestions to improve supervision of monopolistic behaviors among Internet platforms are proposed.

## 1. Introduction

Monopolies involving Internet platforms are a recurring problem. In March 2020, France fined Apple 1.1 billion euros for violating antitrust laws. In April 2021, the State Administration for Market Regulation in China imposed a fine of 18.228 billion yuan on Alibaba Group Holding Co., Ltd. for illegal monopolistic behaviors that prevented merchants from opening stores or engaging in promotional activities on competing platforms. In November 2021, the

**Data Availability Statement:** All relevant data are within the manuscript.

**Funding:** The author(s) received no specific funding for this work.

**Competing interests:** The authors declare that there is no competing interest.

Court of Justice of the European Union (EU) dismissed Google's appeal against the European Commission's Google Shopping Decision and upheld a 2.42 billion euro fine of Google. In December 2021, the Italian antitrust regulator fined Amazon 1.128 billion euros for abusing its market dominance. These incidents have seriously affected market order, economic development, public interest, and social stability. Thus, determining approaches to implementing corresponding anti-monopoly supervision has become an urgent global issue.

In response to antitrust incidents involving Internet platforms, the United States has tightened previously relaxed policies on the platform economy in 2019(www.justice.com). In addition, the Sherman Act of 1890 and Clayton Act of 1914 have been expanded to provide legal support for regulating monopolies in the platform economy, and the Department of Justice has conducted extensive anti-monopoly investigations. The EU in 2021 has strict antitrust laws targeting the platform economy, preventing monopolistic agreements and behaviors among Internet platforms, formulating ex-ante regulations to resolve competition issues in specific industries, and compensating for the lack of supervision covered under existing antitrust laws (An official website of the European Union). In Germany, the Act Against Unfair Competition has been revised and expanded to include authority for anti-monopoly agencies to regulate platform enterprises more effectively in 2021. In Japan, although anti-monopoly laws are generally regarded as a legal norm in 2019, dedicated laws have been formulated targeting the distinct monopoly characteristics common in the platform economy to form a system of collaborative regulation between general and dedicated laws. The Guidelines Concerning Abuse of a Superior Bargaining Position in Transactions between Digital Platform Operators and Consumers that Provide Personal Information, etc. provide a clear definition of superior bargaining position and outline behaviors that are considered abuse of superior bargaining positions. The Fair Online Platform Intermediary Transactions Act, drafted by the Korea Fair Trade Commission, requires Internet platforms to provide a written contract that specifies the terms directly related to sellers' interests during intermediary transactions in March 2021. In South Africa, a competition law enforcement system has been enforced consisting of three agencies (the Competition Tribunal, which adjudicates competition matters; the Competition Commission, which conducts comprehensive anti-monopoly investigations on platform companies; and the Competition Appeal Court) in November 18, 2021. In China, the National Anti-Monopoly Bureau has been established with evolving functions to suit development needs in 2021. China has also focused on improving regulatory systems and mechanisms, innovating supervision concepts and methods, and developing highly knowledgeable and professional teams to enhance its anti-monopoly regulatory capabilities(www.gov.cn).

Although the Chinese government maintains the lead in supervising anti-monopoly practices among Internet platforms, and serves as a leader in establishing and improving the regulatory system, the active cooperation of companies within industry is required. With the development of the Internet, new media has played an important role in information collection, and dissemination and supervision via public opinion has become important for anti-monopoly supervision in the platform economy. Thus, we can argue that the anti-monopoly supervision of Internet platforms requires the joint engagement of the government, companies, new media, and the public.

Therefore, this study aims to construct an evolutionary game theory model involving the above four players within the new media environment context to explore the stable equilibrium point of their strategy selection, analyze the stability of their strategy combination, and the impact of changes in decision variables on strategy selection. We seek to answer the following three questions. (1) How is the strategy selection of the government, Internet platforms, and the public affected by the new media environment? (2) How does the strategy selection of

the government affect the behaviors of Internet platforms, new media, and the public during anti-monopoly co-governance and supervision? (3) How does participation of new media in anti-monopoly co-governance influence the strategy selection of the government, Internet platforms, and the public?

## 2. Literature review

Antitrust issues caused by Internet platforms have attracted widespread attention across all domains and has become a focus of discussion around the world [1]. In recent years, several scholars have conducted extensive research from the perspectives of governments, Internet platforms, new media, and public participation, which collectively provide a foundation for this study.

Despite the leading role played by the government in the anti-monopoly supervision of Internet platforms, some regulatory policies implemented by local governments do not facilitate a timely and tough response against monopolies [2].Therefore, national governments are left with the task of improving the supervision system, preventing disorderly expansion of capital, protecting fair market competition and the legitimate rights and interests of consumers [3], promoting the development of market players, and ensuring healthy competition and business environments [4]. To meet this growing list of governance requirements, governments need innovative measures to supplement the traditional governance model [5]. A focus on regulation by co-governance and supervision [6] can enhance market trust, protect consumer choice and public interest, and tackle the challenges in anti-monopoly supervision caused by the rapid development of Internet platforms [7].

The unique characteristics of the strategic role on the digital market played by large-scale Internet platforms [8] have posed great challenges to anti-monopoly supervision bodies [9]. The supervision system requires further adapting to meet corresponding needs. The monopolistic behavior of Internet platforms has disrupted the market, caused unfair competitive exclusion [10], affected fair competition in the market [11], and hindered the innovative development of enterprises [12]. Hence, it remains essential to strengthen anti-monopoly supervision, which in turn helps to ensure the healthy development, innovation, and vitality of the market.

Information on new media disseminates quickly and has a wide impact [13]. With the rapid development of new media [14], it is essential to understand its impact on Internet platform companies [15]. Information dissemination on new media can either increase the reputation value of these companies or impair it [16], thus having a profound impact [17] and affecting management decisions [18]. At the same time, the government is able to utilize new media [19] to improve efficiency, resolve problems in a timely manner, protect public rights and interests [20], and enhance public support and trust.

Members of the public can express themselves freely through social media platforms and thus, can object to and report the monopoly behaviors of Internet platform companies [21], which forces these companies to improve service [22] and product quality [23] and pay more attention to public influence on corporate decision-making [24]. Effective government supervision also increases the likelihood that Internet platform companies will prioritize public welfare and interests [25], which, in turn, increases public satisfaction with the government. Therefore, the information shared by the public through new media helps anti-monopoly supervision bodies within the government and ensures that Internet platforms treat the attitudes and interests of the public seriously [26].

In summary, existing studies have mainly discussed the roles of the government, Internet platforms, new media, and the public in anti-monopoly supervision from a single player's

perspective. To date, no research has been conducted on the interactive roles of all four players concurrently. In addition, the influence of new media participation on the mutual impact of players' strategy selection during the co-governance and supervision process has yet to be considered.

This study aims to fill the aforementioned research gaps by (1) constructing a game theory model that covers anti-monopoly supervision and includes the government, Internet platforms, new media, and the public as players; (2) exploring the impact of new media participation on players' strategy selection and the combinations of evolutionary stable strategies with and without new media; (3) analyzing the impact of each decision variable on players' strategy selection; and (4) verifying the effectiveness of the model through simulation analysis, using MATLAB 2021b. Countermeasures and suggestions supporting anti-monopoly co-governance and supervision of Internet platform companies are proposed based on the findings.

## 3. Model assumptions and construction

This paper chooses evolutionary game [27, 28] as the research method because it can explain the dynamic process of the evolution of each stakeholder's strategic choice and explain why this state has been reached and how to reach [29]. A conceptual model of the anti-monopoly co-governance and supervision system, involving the government, Internet platforms, new media, and the public, is presented in Fig 1.

### 3.1 Model assumptions

The assumptions of the proposed model are as follows:

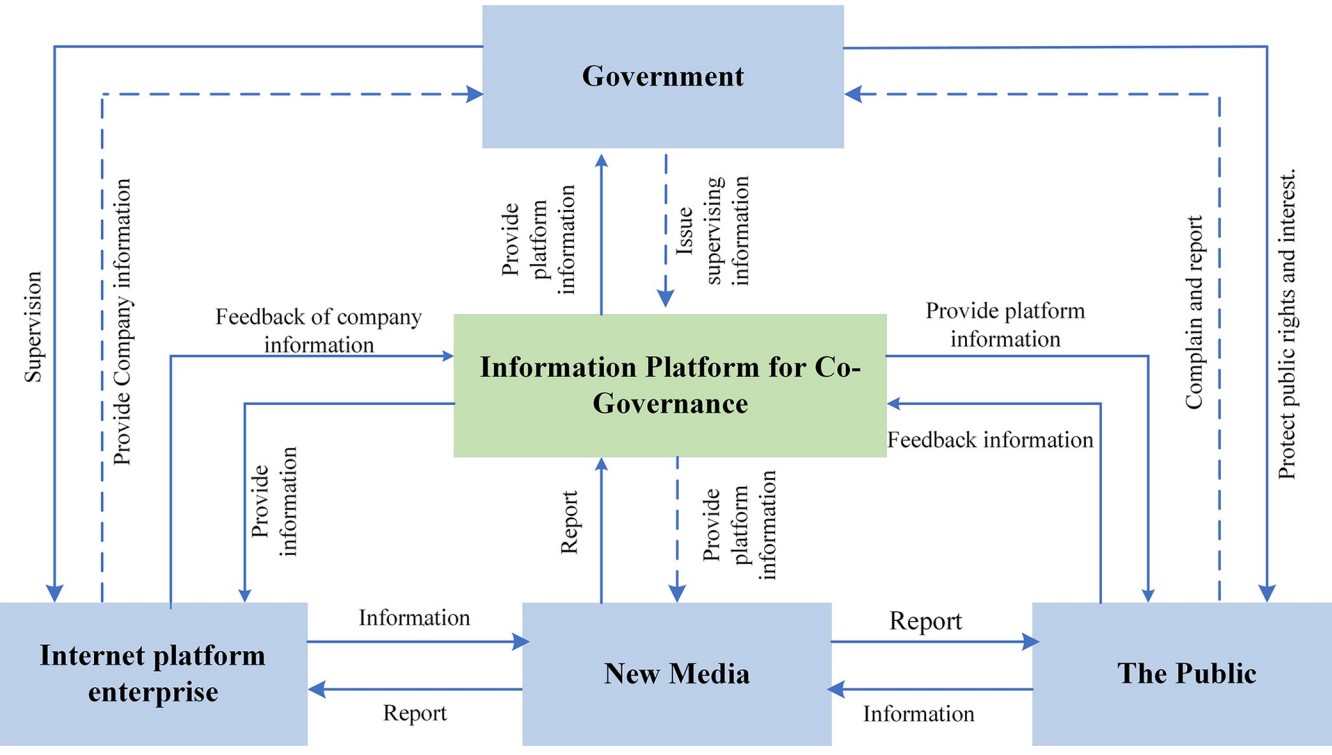

**Fig 1. Four-party anti-monopoly co-governance and supervision system.**

Assumption 1: The strategy selection of the government ($S_g$) is co-governance and supervision or traditional supervision; that of Internet platforms ($S_e$) is operating in compliance with rules and regulations or operating as a monopoly; that of new media ($S_m$) is participating in co-governance or not participating in co-governance; and that of the public ($S_p$) is making a complaint/report or not making a complaint/report. The probabilities of the government choosing co-governance and supervision and traditional supervision are $x$ and $1-x$, respectively; the probabilities of Internet platforms choosing to operate in compliance with the rules and regulations and operating as a monopoly are $y$. and $1-y$, respectively; the probabilities of new media choosing to participate in co-governance and not participating in co-governance are $z$ and $1-z$, respectively; and the probability of the public making a complaint/report and not making a complaint/report are $w$ and $1-w$, respectively. In addition, $x,y,z,w \in [0,1]$.

Assumption 2: The cost of traditional supervision is $C_{gl}$, and the ability to identify monopolistic behavior is $\alpha$, while the cost of co-governance and supervision is $C_{gh}$ ($C_{gh} > C_{gl}$). Using the co-governance and supervision platform improves the ability to identify monopolistic behavior, allows requisition of more valuable information, and promotes public trust in the government, whereby the improved identification ability is $\Delta\alpha$, the information value is $V_g$, and public trust is $H_g$. When the government fails to identify the monopolistic behaviors of Internet platforms, the public's rights and interests are negatively impacted and public trust declines, leading to loss for the government ($D_g$).

Assumption 3: The cost of operating in compliance with rules and regulations is $C_{eh}$, while the cost of operating as a monopoly is $C_{el}$ ($C_{el} < C_{eh}$); the income when operating in compliance with rules and regulations is $R_{el}$, while income when operating as a monopoly is $R_{eh}$ ($R_{eh} > R_{el}$). When the government identifies monopolistic behavior, a fine is imposed on that company as a penalty of the amount $F_e$.

Assumption 4: Participating in co-governance provides new media with more usable information. The cost of participating in co-governance is $C_m$. When Internet platforms operate in compliance with the rules and regulations, the publicity of such information on new media increases their reputation value by $I_e$; however, having monopolistic behavior exposed by new media reduces their reputation value by $N_e$.

Assumption 5: The cost of making a complaint/report is $C_p$. When the government fails to identify the monopolistic behavior of Internet platforms, public rights and interests are damaged at a cost of $L_p$. When new media participates in co-governance, the public is encouraged to make complaints/reports to the new media and, in turn, receives reward $R_p$.

The relevant parameters and descriptions are shown in Table 1.

## 3.2 Constructing the model

The four-party matrix of the game, based on the above assumptions, is exhibited in Table 2.

## 4. Stability of players' strategies

### 4.1 Stability of the government's strategies

The expected payoff of choosing co-governance and supervision is:

$$E_x = yzw(-C_{gh}) + y(1-z)w(-C_{gh}) + (1-y)zw[V_g + H_g - C_{gh} - (1-\alpha-\Delta\alpha)D_g]$$

$$+ (1-y)(1-z)w[V_g + H_g - C_{gh} - (1-\alpha-\Delta\alpha)D_g] + yz(1-w)(-C_{gh}) + y(1-z)(1-w) \times (-C_{gh})$$

**Table 1. Parameters and descriptions.**

| Parameter | Description | Parameter | Description |
|---|---|---|---|
| $x$ | Probability of the government preferring co-governance and supervision | $C_{eh}$ | Cost of Internet platforms when operating in compliance with rules and regulations |
| $y$ | Probability of Internet platforms operating in compliance with the rules and regulations | $C_{el}$ | Cost of Internet platforms when operating as a monopoly |
| $z$ | Probability of new media participating in co-governance | $R_{el}$ | Income of Internet platforms when operating in compliance with rules and regulations |
| w | Probability of the public making a complaint/report | $R_{eh}$ | Income of Internet platforms when operating as monopoly |
| $C_{gl}$ | Cost for the government applying traditional supervision | $F_e$ | Fine amount imposed on Internet platforms when operating as monopoly |
| $C_{gh}$ | Cost for the government applying co-governance and supervision | $C_m$ | Cost of new media participating in co-governance |
| $\alpha$ | The government's ability to identify monopolistic behavior using traditional supervision | $V_m$ | Information value gained by new media when participating in co-governance |
| $V_g$ | Information value gained by the government when using the co-governance and supervision platform | $I_e$ | Increase in Internet platforms' reputation value when positive information is posted on new media |
| $H_g$ | The increase in public trust towards the government | $N_e$ | Reduction in Internet platforms' reputation value when monopolistic behavior is exposed on new media |
| $D_g$ | The loss for the government when failing to identify monopolistic behavior | $C_p$ | Cost to the public when making a complaint/report |
| $L_p$ | The cost from the damage to public rights and interests caused by Internet platforms' monopolistic behaviors | $R_p$ | Reward given to the public by new media for making complaints/report |

$$+(1-y)z(1-w)[V_g + H_g - C_{gh} - (1-\alpha-\Delta\alpha)D_g] + (1-y)(1-z)(1-w)[V_g + H_g - C_{gh} - (1-\alpha-\Delta\alpha)D_g]$$

$$= -C_{gh} + (1-y)[V_g + H_g - (1-\alpha-\Delta\alpha)D_g] \tag{1}$$

The expected payoff of choosing traditional supervision is:

$$E_{1-x} = yzw(-C_{gl}) + y(1-z)w(-C_{gl}) + (1-y)zw[-C_{gl} - (1-\alpha)D]_g$$

$$+(1-y)(1-z)w[-C_{gl} - (1-\alpha)D_g] + yz(1-w)(-C_{gl}) + y(1-z)(1-w)(-C_{gl})$$

$$+(1-y)z(1-w)[-C_{gl} - (1-\alpha)D_g] + (1-y)(1-z)(1-w)[-C_{gl} - (1-\alpha)D_g]$$

$$= -C_{gl} + (1-y)[-(1-\alpha)D_g] \tag{2}$$

The replicator dynamics equation and first derivative of the government's strategy selection are

$$F(x) = dg/dt = x(E_x - \bar{E}) = x(1-x)(E_x - E_{1-x})$$

$$= x(1-x)\{-C_{gh} + C_{gl} + (1-y)[V_g + H_g + \Delta\alpha D_g]\} \tag{3}$$

$$F'(x) = (1-2x)\{-C_{gh} + C_{gl} + (1-y)[V_g + H_g + \Delta\alpha D_g]\} \tag{4}$$

**Table 2. Four-party matrix.**

| Internet platforms | New Media | Public | Government | | | |
|---|---|---|---|---|---|---|
| | | | **Traditional supervision** **1−x** | | **Co-governance and supervision** **x** | |
| | | | **Make a complaint/report** **w** | **Not make a complaint/report** **1−w** | **Make a complaint/report** **w** | **Not make a complaint/report** **1−w** |
| **Operate in compliance with rules and regulations $y$** | **Participate in co-governance $z$** | | $-C_{gl}$ $-C_p$ $-C_m$ $R_{el}-C_{eh}+I_e$ | $-C_{gl}$ $0$ $-C_m$ $R_{el}-C_{eh}+I_e$ | $-C_{gh}$ $-C_p$ $-C_m$ $R_{el}-C_{eh}+I_e$ | $-C_{gh}$ $0$ $-C_m$ $R_{el}-C_{eh}+I_e$ |
| | **Not participate in co-governance $1−z$** | | $-C_{gl}$ $-C_p$ $0$ $R_{el}-C_{eh}$ | $-C_{gl}$ $0$ $0$ $R_{el}-C_{eh}$ | $-C_{gh}$ $-C_p$ $0$ $R_{el}-C_{eh}$ | $-C_{gh}$ $0$ $0$ $R_{el}-C_{eh}$ |
| **Operate as a monopoly $1−y$** | **Participate in co-governance $z$** | | $-C_{gl}-(1-\alpha)D_g$ $R_p-C_p-(1-\alpha)L_p$ $-C_m+V_m-R_p$ $R_{eh}-C_{el}-\alpha(F_e+N_e)$ | $-C_{gl}-(1-\alpha)D_g$ $-(1-\alpha)L_p$ $-C_m$ $R_{eh}-C_{el}-\alpha(F_e+N_e)$ | $V_g+H_g-C_{gh}-(1-\alpha-\Delta\alpha)D_g$ $R_p-C_p-(1-\alpha-\Delta\alpha)L_p$ $-C_m+V_m-R_p$ $R_{eh}-C_{el}-(\alpha+\Delta\alpha)(F_e+N_e)$ | $V_g+H_g-C_{gh}-(1-\alpha-\Delta\alpha)D_g$ $-(1-\alpha-\Delta\alpha)L_p$ $-C_m$ $R_{eh}-C_{el}-(\alpha+\Delta\alpha)(F_e+N_e)$ |
| | **Not participate in co-governance $1−z$** | | $-C_{gl}-(1-\alpha)D_g$ $-C_p-(1-\alpha)L_p$ $0$ $R_{eh}-C_{el}-\alpha F_e$ | $-C_{gl}-(1-\alpha)D_g$ $-(1-\alpha)L_p$ $0$ $R_{eh}-C_{el}-\alpha F_e$ | $V_g+H_g-C_{gh}-(1-\alpha-\Delta\alpha)D_g$ $-C_p-(1-\alpha-\Delta\alpha)L_p$ $0$ $R_{eh}-C_{el}-(\alpha+\Delta\alpha)F_e$ | $V_g+H_g-C_{gh}-(1-\alpha-\Delta\alpha)D_g$ $-(1-\alpha-\Delta\alpha)L_p$ $0$ $R_{eh}-C_{el}-(\alpha+\Delta\alpha)F_e$ |

According to the stability theory of differential equations, for a probability (in this case, the probability of choosing co-governance and supervision) to maintain a stable equilibrium, the equations must satisfy $F(x) = 0$ and $F'(x)<0$.

**Proposition 1:** When $y<y_0$, the stable strategy of the government is co-governance and supervision; when $y>y_0$, the stable strategy is traditional supervision; and when $y = y_0$, a stable strategy cannot be determined. In addition, the threshold $y_0 = (-C_{gh} + C_{gl})/(V_g + H_g + \Delta\alpha D_g) + 1$.

**Proof:** Let $G(y, \Delta\alpha) = -C_{gh} + C_{gl} + (1 - y)[V_g + H_g + \Delta\alpha D_g]$, when $G(y,\Delta\alpha) = 0$, $y_0 = 1 - (C_{gh} - C_{gl})/(V_g + H_g + \Delta\alpha D_g)$ and $\Delta\alpha_0 = [(C_{gh} - C_{gl})/(1 - y) - V_g - H_g]/D_g$, because $\partial G(y,\Delta\alpha)/\partial y < 0$ and $\partial G(y,\Delta\alpha)/\partial\Delta\alpha > 0$, $G(y,\Delta\alpha)$ is a decreasing function of $y$ and an increasing function of $\Delta\alpha$. When $y<y_0$ or $\Delta\alpha>\Delta\alpha_0$, $G(y,\Delta\alpha)>0$, $F'(x)|_{x=1} < 0$, $F(x)|_{x=1} = 0$, and the equilibrium solution $x = 1$ is stable. In the same way, when $y>y_0$ or $\Delta\alpha<\Delta\alpha_0$, $G(y,\Delta\alpha)<0$, $F'(x)|_{x=0} < 0$, $F(x)|_{x=0} = 0$, and the equilibrium solution $x = 0$ is stable. However, when $y = y_0$ or $\Delta\alpha = \Delta\alpha_0$, $G(y,\Delta\alpha) = 0$, $F'(x) = 0$, and a stable strategy cannot be determined.

Thus, the response function for the probability of the government choosing co-governance and supervision ($x$) is

$$x = \begin{cases} 0 & if \quad y > y_0 \quad or \quad \Delta\alpha < \Delta\alpha_0 \\ (0, 1) & if \quad y = y_0 \quad or \quad \Delta\alpha = \Delta\alpha_0 \\ 1 & if \quad y < y_0 \quad or \quad \Delta\alpha > \Delta\alpha_0 \end{cases} \quad (5)$$

Proposition 1 shows that, when the probability of Internet platform companies' operating in compliance with rules and regulations is low or when the government's ability to identify monopolistic behavior increases, the probability of the government preferring co-governance

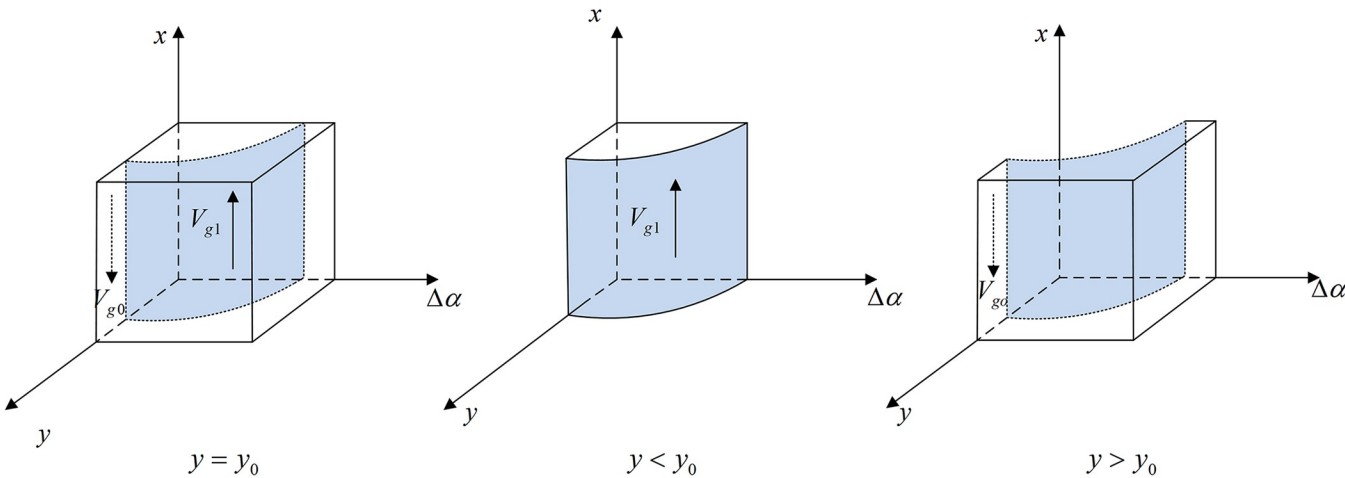

**Fig 2. Phase diagram of the government's strategy selection.**

and supervision tends to stabilize at 1; conversely, when the probability of Internet platform companies' operating in compliance with rules and regulations is high or when the government's ability to identify monopolistic behavior decreases, the probability of the government choosing co-governance and supervision tends to stabilize at 0. Therefore, when Internet platform companies show monopolistic behaviors, the government chooses a strategy of co-governance and supervision to improve regulatory efficiency.

The phase diagram of the government's strategy selection is shown in Fig 2.

Fig 2 shows that the volume of $V_{g1}$ demonstrates the probability of the government choosing the co-governance and supervision strategy, and the volume of $V_{g0}$ demonstrates the probability of choosing traditional supervision. In addition,

$$V_{x1} = \int_0^1 \int_0^1 1 - \frac{C_{gh} - C_{gl}}{V_g + H_g + \Delta\alpha D_g} d\Delta\alpha dx = 1 - \frac{C_{gh} - C_{gl}}{D_g} \ln\frac{V_g + H_g + \Delta\alpha D_g}{V_g + H_g} \quad (6)$$

$$V_{x0} = 1 - V_{x1} = \frac{C_{gh} - C_{gl}}{D_g} \ln\frac{V_g + H_g + \Delta\alpha D_g}{V_g + H_g} \quad (7)$$

**Corollary 1.1:** The probability of the government choosing the traditional supervision strategy is positively correlated to the costs of co-governance and supervision, and negatively correlated to loss of the negative impact caused by failing to identify monopolistic behavior.

**Proof:** Based on $V_{x0}$, the first-order partial derivatives of $C_{gh}$ and $D_g$ can be obtained as follows:

$$\frac{\partial V_{x0}}{\partial C_{gh}} = \frac{1}{D_g} \ln\frac{V_g + H_g + \Delta\alpha D_g}{V_g + H_g} > 0$$

$$\frac{\partial V_{x0}}{\partial D_g} = -\left[\frac{C_{gh} - C_{gl}}{D_g^2} \ln\frac{V_g + H_g + \Delta\alpha D_g}{V_g + H_g} + \frac{C_{gh} - C_{gl}}{V_g + H_g + \Delta\alpha D_g}\right] < 0$$

End of proof.

Corollary 1.1 shows that, when the cost of co-governance and supervision is large, the probability of the government choosing the traditional supervision strategy is also large, while, when choosing traditional supervision results in increased losses, the probability of choosing traditional supervision declines.

**Corollary 1.2:** The probability of the government choosing the co-governance and supervision strategy is positively correlated to the costs of applying traditional supervision, as well as the information value obtained from the co-governance and supervision platform and the increase in public trust.

**Proof:** Based on $V_{x1}$, the first-order partial derivatives of $C_{gl}$, $V_g$, and $H_g$ can be obtained as follows:

$$\frac{\partial V_{x1}}{\partial C_{gl}} = \frac{1}{D_g} \ln \frac{V_g + H_g + \Delta\alpha D_g}{V_g + H_g} > 0$$

$$\frac{\partial V_{x1}}{\partial V_g} = \frac{(C_{gh} - C_{gl})\Delta\alpha}{(V_g + H_g + \Delta\alpha D_g)(V_g + H_g)} > 0$$

$$\frac{\partial V_{x1}}{\partial H_g} = \frac{(C_{gh} - C_{gl})\Delta\alpha}{(V_g + H_g + \Delta\alpha D_g)(V_g + H_g)} > 0$$

End of proof.

Corollary 1.2 shows that increase in the cost of traditional supervision, information value obtained from the co-governance platform, and public trust make the government more inclined to choose the co-governance and supervision strategy.

**Corollary 1.3**: When the government's ability to identify monopolistic behavior improves ($\Delta\alpha > \Delta\alpha_0$), the government chooses co-governance and supervision; however, when $\Delta\alpha < \Delta\alpha_0$, the government chooses traditional supervision. In addition, the threshold $\Delta\alpha_0 = [C_{gh} - C_{gl}/(1-y) - V_g - H_g]/D_g$.

**Proof:** According to Proposition 1, when $G(y,\Delta\alpha) = 0$, $\Delta\alpha_0 = [C_{gh} - C_{gl}/(1-y) - V_g - H_g]/D_g$. Because $\partial G(y,\Delta\alpha)/\partial\Delta\alpha > 0$, $G(y,\Delta\alpha)$ is an increasing function of $\Delta\alpha$. When $\Delta\alpha > \Delta\alpha_0$, $G(y,\Delta\alpha) > 0$, then $F'(x)|_{x=1} < 0$, whilst when $\Delta\alpha < \Delta\alpha$, $G(y,\Delta\alpha) < 0$, then $F'(x)|_{x=0} < 0$. End of proof.

Corollary 1.3 shows that the greater government's ability to identify monopolistic behavior through utilizing the co-governance platform, the more likely it is to choose co-governance and supervision. When that identification ability drops below the threshold ($\Delta\alpha$), the government chooses traditional supervision. Therefore, improving the effectiveness of the co-governance platform when identifying monopolistic behavior promotes the chances of selecting the co-governance and supervision strategy.

## 4.2 Stability of internet platforms' strategies

The expected payoff of choosing to operate in compliance with the rules and regulations is

$$E_y = (1-x)zw(R_{el} - C_{eh} + I_e) + (1-x)z(1-w)(R_{el} - C_{eh} + I_e)$$

$$+ xzw(R_{el} - C_{eh} + I_e) + xz(1-w)(R_{el} - C_{eh} + I_e)$$

$$+ (1-x)(1-z)w(R_{el} - C_{eh}) + (1-x)(1-z)(1-w)(R_{el} - C_{eh})$$

$$+ x(1-z)w(R_{el} - C_{eh}) + x(1-z)(1-w)(R_{el} - C_{eh})$$

$$= R_{el} - C_{eh} + zI_e \tag{8}$$

The expected payoff of choosing to operate as a monopoly is

$$E_{1-y} = (1-x)zw[R_{eh} - C_{el} - \alpha(F_e + N_e)] + (1-x)z(1-w)[R_{eh} - C_{el} - \alpha(F_e + N_e)]$$

$$+ xzw[R_{eh} - C_{el} - (\alpha + \Delta\alpha)(F_e + N_e)] + xz(1-w)[R_{eh} - C_{el} - (\alpha + \Delta\alpha)(F_e + N_e)]$$

$$+ (1-x)(1-z)w(R_{eh} - C_{el} - \alpha F_e) + (1-x)(1-z)(1-w)(R_{eh} - C_{el} - \alpha F_e)$$

$$+ x(1-z)w[R_{eh} - C_{el} - (\alpha + \Delta\alpha)F_e] + x(1-z)(1-w)[R_{eh} - C_{el} - (\alpha + \Delta\alpha)F_e]$$

$$= R_{eh} - C_{el} - \alpha F_e - x\Delta\alpha F_e - z\alpha N_e - zx\Delta\alpha N_e \tag{9}$$

The replicator dynamics equation and first derivative of Internet platforms' strategy selection are

$$F(y) = dy/dt = y(E_e - \bar{E}) = y(1-y)(E_e - E_{1-e})$$

$$= y(1-y)(R_{el} - R_{eh} + C_{el} - C_{eh} + zI_e + \alpha F_e + x\Delta\alpha F_e + z\alpha N_e + zx\Delta\alpha N_e) \tag{10}$$

$$F'(y) = (1-2y)[(R_{el} - R_{eh} + C_{el} - C_{eh} + zI_e + \alpha F_e + x\Delta\alpha F_e + z\alpha N_e + zx\Delta\alpha N_e)] \tag{11}$$

According to stability theory, the probability of the choice to operate in compliance with the rules and regulations to maintain a stable state, the equations must satisfy $F(y) = 0$ and $F'(y) < 0$.

**Proposition 2:** When $x > x_0$ or $z > z_0$, the stable strategy of Internet platforms operates in compliance with rules and regulations; when $x < x_0$ or $z < z_0$, the stable strategy operates as a monopoly; when $x = x_0$ or $z = z_0$, the stable strategy cannot be determined. In addition, the threshold $x_0 = R_{eh} - R_{el} + C_{eh} - C_{el} - zI_e - \alpha F - z\alpha N_e / (\Delta\alpha F_e + z\Delta\alpha N_e)$,
$z_0 = R_{eh} - R_{el} + C_{eh} - C_{el} - \alpha F_e - x\Delta\alpha F_e - x\Delta\alpha N_e / (I_e + \alpha N_e)$.

Proof: Let $G(x,z) = R_{el} - R_{eh} + C_{el} - C_{eh} + zI_e + \alpha F_e + x\Delta\alpha F_e + z\alpha N_e + x\Delta\alpha N_e$. When $G(x,z) = 0$, $x_0 = \frac{R_{eh} - R_{el} + C_{eh} - C_{el} - zI_e - \alpha F - z\alpha N_e}{\Delta\alpha F_e + z\Delta\alpha N_e}$, and $z_0 = \frac{R_{eh} - R_{el} + C_{eh} - C_{el} - \alpha F_e - x\Delta\alpha F_e - x\Delta\alpha N_e}{(I_e + \alpha N_e)}$.

Since $\partial G(x,z)/\partial x < 0$, $\partial G(x,z)/\partial z < 0$, $G(x,z)$ is a decreasing function of $x$ and $z$.

when $x < x_0$ or $z < z_0$, $G(x,z) > 0$, $F'(y)|_{y=1} < 0$, $F(y)|_{y=1} = 0$, and the equilibrium solution $y = 1$ is stable. In the same way, when $x > x_0$ or $z > z_0$, $G(x,z) < 0$, $F'(y)|_{y=0} < 0$, $F(y)|_{y=0} = 0$, and the equilibrium solution $y = 0$ is stable. However, when $x = x_0$ or $z = z_0$, $G(x,z) = 0$, $F'(y) = 0$, and a stable strategy cannot be determined.

Thus, the response function for the probability of Internet platforms choosing to operate in compliance with rules and regulations ($y$) is

$$y = \begin{cases} 0 & if \quad x > x_0 \quad or \quad z > z_0 \\ (0,1) & if \quad x = x_0 \quad or \quad z = z_0 \\ 1 & if \quad x < x_0 \quad or \quad z < z_0 \end{cases} \tag{12}$$

Proposition 2 shows that, when the probability of the government choosing the co-governance and supervision strategy is low, the stable strategy of Internet platforms is to operate as a monopoly, and when the probability of the government choosing the co-governance and supervision strategy increases, the stability strategy shifts to operate in compliance with rules and regulations. When the probability of new media choosing to participate in co-governance is high, the stable strategy is to operate in compliance with rules and regulations. When the government actively applies co-governance and supervision, Internet platforms are more likely to choose to operate in compliance with rules and regulations. Hence, new media participation in co-governance can increase the probability of Internet platforms choosing to operate in compliance with rules and regulations.

Fig 3 shows the phase diagram of Internet platforms' strategy selection.

Fig 3 shows that the volume of $V_{y1}$ demonstrates the probability of Internet platforms choosing to operate in compliance with rules and regulations, and the volume of $V_{y0}$ demonstrates the Internet platforms' choice to operate as a monopoly. In addition,

$$V_{y0} = \int_0^1 \int_0^1 \frac{R_{eh} - R_{el} + C_{eh} - C_{el} - zI_e - \alpha F_e - z\alpha N_e}{\Delta\alpha F_e + z\Delta\alpha N_e} dzdy$$

$$= \frac{R_{eh} - R_{el} + C_{eh} - C_{el} - \alpha F_e - 0.5I_e - 0.5\alpha N_e}{\Delta\alpha N_e} \ln(\Delta\alpha F_e + \Delta\alpha N_e) \tag{13}$$

$$V_{y1} = 1 - V_{y0} = 1 - \frac{R_{eh} - R_{el} + C_{eh} - C_{el} - \alpha F - 0.5I_e - 0.5\alpha N_e}{\Delta\alpha N_e} \ln(\Delta\alpha F_e + \Delta\alpha N_e) \tag{14}$$

**Corollary 2.1:** The probability of Internet platforms choosing to operate as a monopoly is positively correlated with the additional income generated through monopolistic operations,

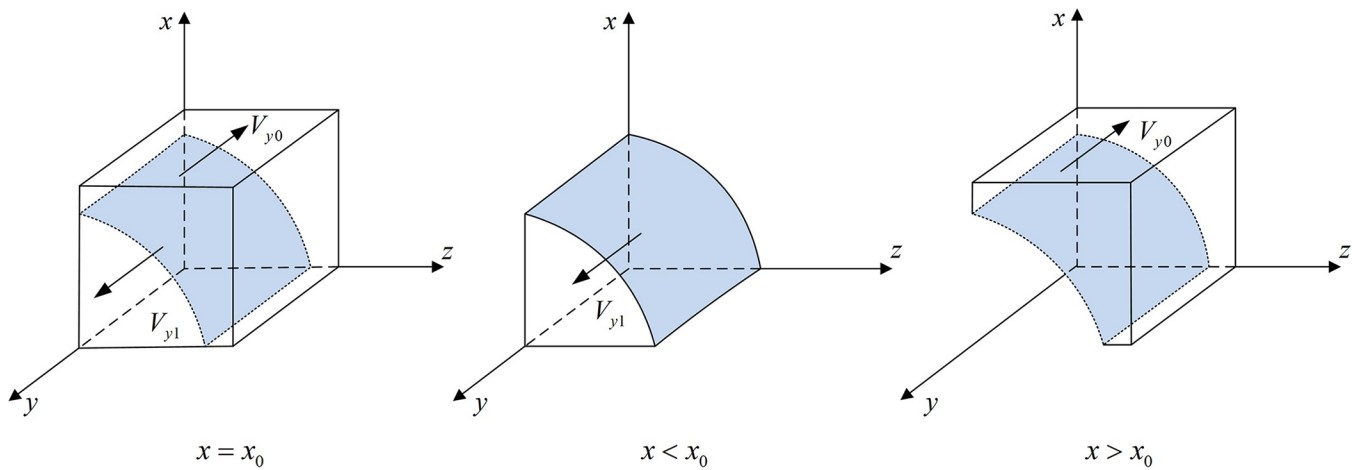

**Fig 3. Phase diagram of internet platforms' strategy selection.**

and negatively correlated with the fine imposed by the government when they are caught, as well as the loss in reputation value caused by exposure to such behavior on new media.

**Proof:** Based on $V_{y0}$, the first-order partial derivatives of $(R_{eh}-R_{el})$, $F_e$, and $N_e$ can be obtained as follows:

$$\frac{\partial V_{y0}}{\partial(R_{eh}-R_{el})} = \frac{1}{\Delta\alpha N_e}\ln(\Delta\alpha F_e + \Delta\alpha N_e) > 0$$

$$\frac{\partial V_{e0}}{\partial F_e} = -\frac{\alpha}{\Delta\alpha N_e}\ln(\Delta\alpha F_e + \Delta\alpha N_e) < 0$$

$$\frac{\partial V_{e0}}{\partial N_e} = -\frac{\Delta\alpha^2(R_{eh}-R_{el}+C_{eh}-C_{el}-\alpha F_e-0.5I_e)}{\Delta\alpha^2 N_e^2 F_e + \Delta\alpha^2 N_e^3} < 0$$

End of proof.

Corollary 2.1 shows that the higher income from operating as a monopoly than operating in compliance with the rules and regulations, the more likely that Internet platforms are to choose monopolistic operations. However, the greater the penalty imposed by the government, the less likely they are to choose monopolistic operations. In addition, a greater reputation loss discourages monopolistic behavior.

**Corollary 2.2:** The probability that Internet platforms choose to operate in compliance with the rules and regulations is positively correlated to the additional costs of compliance operation as well as the increase in reputation value generated by positive publicity in the new media.

**Proof:** Based on $V_{y1}$, the first-order partial derivatives of $(C_{eh}-C_{el})$ and $I_e$ can be obtained as follows:

$$\frac{\partial V_{y0}}{\partial(C_{eh}-C_{el})} = \frac{1}{\Delta\alpha N_e}\ln(\Delta\alpha F_e + \Delta\alpha N_e) > 0$$

$$\frac{\partial V_{e0}}{\partial I_e} = -\frac{\alpha}{\Delta\alpha N_e}\ln(\Delta\alpha F_e + \Delta\alpha N_e) < 0$$

End of proof.

Corollary 2.2 shows that the higher the additional costs involved in operating in compliance with the rules and regulations, the more likely companies will operate as a monopoly. However, when the penalty for operating as a monopoly is large, they are less likely to choose to operate monopolistically.

### 4.3 Stability of new media strategies

The expected payoff of choosing to participate in co-governance is

$$E_z = (1-x)yw(-C_m) + (1-x)y(1-w)(-C_m) + xyw(-C_m) + xy(1-w)(-C_m)$$

$$+(1-x)(1-y)w(-C_m + V_m - R_p) + (1-x)(1-y)(1-w)(-C_m)$$

$$+x(1-y)w(-C_m + V_m - R_p) + x(1-y)(1-w)(-C_m)$$

$$= -C_m + w(1-y)(V_m - R_p) \tag{15}$$

The expected payoff of choosing not to participate in co-governance is

$$E_{1-z} = (1-x)yw*0 + (1-x)y(1-w)*0 + xyw*0 + xy(1-w)*0 + (1-x)(1-y)w*0$$

$$= +(1-x)(1-y)(1-w)*0 + x(1-y)w*0 + x(1-y)(1-w)*0 = 0 \tag{16}$$

The replicator dynamics equation and first derivative of new media's strategy selection are

$$F(z) = dp/dt = z(E_m - \bar{E}) = z(1-z)(E_{m1} - E_{m2})$$
$$= z(1-z)[-C_m + w(1-y)(V_m - R_p)] \tag{17}$$

$$F'(z) = (1-2z)[-C_m + w(1-y)(V_m - R_p)] \tag{18}$$

Stability theory contends that, for the probability of choosing to participate in co-governance to be in a stable state, the equations must satisfy $F(z) = 0$ and $F'(z){<}0$.

**Proposition 3:** When $y{<}y_3$ or $w{>}w_3$, the stable strategy of new media is participating in co-governance; when $y{>}y_3$ or $w{<}w_3$, the stable strategy is not participating in co-governance; and when $y = y_3$ or $w = w_3$, the stable strategy cannot be determined. In addition, the threshold $y_3 = 1 - C_m/w(V_m - R_p)$, $w_3 = C_m/(1-y)(V_m - R_p)$.

**Proof:** Let $K(y, w) = -C_m + w(1-y)(V_m - R_p)$. When $K(y,w) = 0$, $y_3 = 1-C_m/w(V_m-R_p)$ and $w_3 = C_m/(1-y)(V_m-R_p)$. Because $\partial K(y, w)/\partial y > 0$, $\partial K(y, w)/\partial w > 0$, $K(y,w)$ is an increasing function of $y$ and $w$. When $y{>}y_3$ or $w{>}w_3$, $K(y,w){>}0$, $F'(z)|_{z=1} < 0$, $F(z)|_{z=1} = 0$, and the equilibrium solution $z = 1$ is stable. In the same way, when $y{<}y_3$ or $w{<}w_3$, $K(y,w){<}0$, $F'(z)|_{z=0} < 0$, $F(z)|_{z=0} = 0$, and the equilibrium solution $z = 0$ is stable. However, when $y = y_3$ or $w = w_3$, $K(y,w) = 0$, $F'(z){=}0$, and a stable strategy cannot be determined.

Thus, the response function for the probability of new media choosing to participate in co-governance ($z$) is

$$z = \begin{cases} 0 & if \quad y < y_3 \quad or \quad w < w_3 \\ (0,1) & if \quad y = y_3 \quad or \quad w = w_3 \\ 1 & if \quad y > y_3 \quad or \quad w > w_3 \end{cases} \tag{19}$$

Proposition 3 shows that, as the probability for the public making complaints/reports increases, the stability strategy of new media changes from not participating in co-governance to participating; when the probability of Internet platforms operating in compliance with the rules and regulations is low, the stability strategy of new media also changes from not participating in co-governance to participating. In other words, when public enthusiasm to make complaints/reports is high, or when the probability of Internet platforms operating in

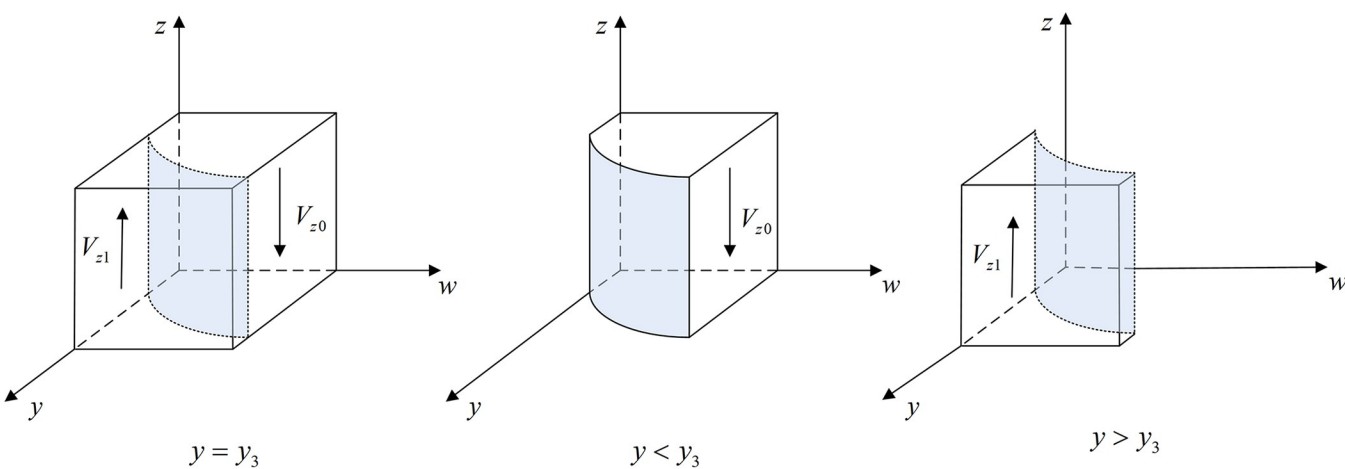

**Fig 4. Phase diagram of new media's strategy selection.**

compliance with rules and regulations is low, new media are more likely to utilize the information provided by the public to participate in the co-governance of monopolies.

Fig 4 shows the phase diagram of new media's strategy selection.

$$V_{z1} = \int_0^1 \int_{\frac{C_m}{V_m - R_p}}^1 1 - \frac{C_m}{w(V_m - R_p)} \, dw \, dz = 1 - \frac{C_m}{V_m - R_p} + \frac{C_m}{V_m - R_p} \ln \frac{C_m}{V_m - R_p} \quad (20)$$

$$V_{z0} = 1 - V_{z1} = \frac{C_m}{V_m - R_p} - \frac{C_m}{V_m - R_p} \ln \frac{C_m}{V_m - R_p} \quad (21)$$

**Corollary 3.1:** The probability of the new media choosing to participate in co-governance is negatively correlated to the cost of such participation and positively correlated to the information value obtained through participation.

**Proof:** Based on $V_{z1}$, the first-order partial derivatives of $C_m$ and $V_m$ can be obtained as follows:

$$\frac{\partial V_{z1}}{\partial C_m} = -\frac{1}{V_m - R_p} \ln \frac{V_m - R_p}{C_m} < 0$$

$$\frac{\partial V_{z1}}{\partial V_m} = \frac{C_m}{(V_m - R_p)^2} \ln \frac{V_m - R_p}{C_m} > 0$$

End of proof.

Corollary 3.1 shows that reducing participation costs or increasing the information value increases the likelihood of new media's participation in co-governance.

**Corollary 3.2:** The probability of new media choosing not to participate in co-governance is positively correlated to the cost of participation and negatively correlated to the rewards that new media give to the public for making complaints/reports.

Proof: Based on $V_{z0}$, the first-order partial derivatives of $C_m$ and $R_p$ can be obtained as follows:

$$\frac{\partial V_{z0}}{\partial C_m} = \frac{1}{V_m - R_p} \ln \frac{V_m - R_p}{C_m} > 0$$

$$\frac{\partial V_{z0}}{\partial R_P} = -\frac{C_m}{(V_m - R_p)^2} \ln \frac{C_m}{V_m - R_p} < 0$$

End of proof.

Corollary 3.2 shows that, when the costs of participation are high, the probability of new media choosing not to participate in co-governance is also high. When new media provides larger rewards to the public for providing information through complaints/reports, the public is more enthusiastic in doing so, which increases the value of the information obtained by new media and stimulates its participation in co-governance strategies.

## 4.4 Stability of the public's strategies

The expected payoff of making a complaint/report is

$$E_{p1} = (1-x)yz(-C_p) + (1-x)y(1-z)(-C_p) + (1-x)(1-y)z[R_p - C_p - (1-\alpha)L_p]$$

$$+(1-x)(1-y)(1-z)[-C_p - (1-\alpha)L_p] + xyz(-C_p) + xy(1-z)(-C_p)$$

$$+x(1-y)z[R_p - C_p - (1-\alpha-\Delta\alpha)L_p] + x(1-y)(1-z)[-C_p - (1-\alpha-\Delta\alpha)L_p]$$

$$= -C_P + z(1-y)R_p - (1-y)(1-\alpha)L_p + (1-y)x\Delta\alpha L_p \tag{22}$$

The expected payoff of not making a complaint/report is

$$E_{p2} = (1-x)(1-y)z[-(1-\alpha)L_p] + (1-x)(1-y)(1-z)[-(1-\alpha)L_p]$$

$$+x(1-y)z[-(1-\alpha-\Delta\alpha)L_p] + x(1-y)(1-z)[-(1-\alpha-\Delta\alpha)L_p]$$

$$= -(1-y)(1-\alpha)L_p + (1-y)x\Delta\alpha L_p \tag{23}$$

The replicator dynamics equation and first derivative of the public's strategy selection are

$$F(w) = dp/dt = w(E_p - \bar{E}) = w(1-w)(E_{p1} - E_{p2}) = w(1-w)[-C_p + z(1-y)R_p] \tag{24}$$

$$F'(w) = (1-2w)[-C_P + z(1-y)R_p] \tag{25}$$

As with previous sections, for the probability of choosing to make a complaint/report to be in a stable state, the equations must satisfy $F(w) = 0$ and $F'(w) < 0$.

**Proposition 3:** When $e > e_2$ or $m > m_2$, the stable strategy of the public is to post a truthful report; when $e < e_2$ or $m < m_2$, the stable strategy of the public is to post a distorted report; and when $e = e_2$ or $m = m_2$, the stable strategy cannot be determined. In addition, the threshold $y_4 = zR_p - C_p/zR_p$, $z_4 = C_p/R_p(1-y)$.

Proof: Let $J(y,z) = -C_p + z(1-y)R_p$. When $J(y,z) = 0$, $y = 1 - \frac{C_p}{zR_p}$ and $z = \frac{C_p}{R_p(1-y)}$. Because $\partial J(y,z)/\partial y < 0$, $\partial J(y,z)/\partial z > 0$, $J(y,z)$ is an increasing function of $y$ and $z$. When $y < y_4$ or

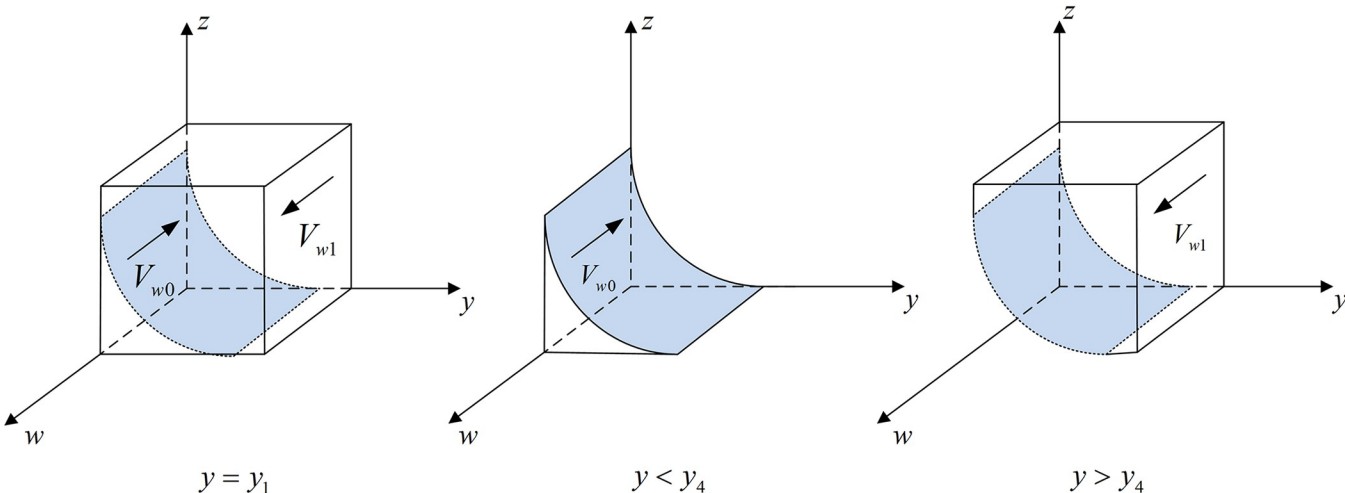

**Fig 5. Phase diagram of new public's strategy selection.**

$z > z_4$, $J(y,z) > 0$, $F'(w)|_{w=1} < 0$, $F(w)|_{w=1} = 0$, and the equilibrium solution $w = 1$ is stable. In the same way, when $y > y_4$ or $z < z_4$, $J(y,z) < 0$, $F'(w)|_{w=0} < 0$, $F(w)|_{w=0} = 0$, and the equilibrium solution $w = 0$ is stable. However, when $y = y_4$ or $z = z_4$, $J(y,z) = 0$, $F'(w) = 0$, and a stable strategy cannot be determined.

Thus, the response function for the probability of the public choosing to make a complaint/report ($w$) is

$$w = \begin{cases} 0 & if \quad y > y_4 \quad or \quad z < z_4 \\ (0,1) & if \quad y = y_4 \quad or \quad z = z_4 \\ 1 & if \quad y < y_4 \quad or \quad z > z_4 \end{cases} \tag{26}$$

Proposition 4 shows that, when the probability of Internet platforms operating in compliance with rules and regulations is high, the public's stable strategy is not to make a complaint/report. As the probability of Internet platform companies' monopolistic operations increases, the public's stability strategy changes from not making a complaint/report to making a complaint/report. When the probability of new media choosing to participate in co-governance is high, the public's stable strategy is to make a complaint/report. In other words, when Internet platforms operate monopolistically, the public is more inclined to make complaints/reports, whilst receiving information from the public on the co-governance platform increases the probability of new media participating in the co-governance.

Fig 5 shows the phase diagram of the public's strategy selection.

Fig 5 shows that the volume of $V_{w1}$ demonstrates the probability of choosing to make a complaint/report, and the volume of $V_{w0}$ demonstrates the probability of choosing not to make a complaint/report. In addition,

$$V_{w0} = \int_0^1 \int_0^{\frac{R_p - C_p}{R_p}} \frac{C_p}{R_{p(1-y)}} dy dw = \frac{C_p}{R_p} \ln \frac{R_p}{C_p} \tag{27}$$

$$V_{w1} = 1 - V_{w0} = 1 - \frac{C_p}{R_p} \ln \frac{R_p}{C_p} \tag{28}$$

**Corollary 4.1:** The probability of the public choosing not to make a complaint/report is positively correlated to the cost of doing so and negatively correlated to the rewards received from new media.

Proof: Based on $V_{w0}$, the first-order partial derivatives of $C_p$ and $R_p$ can be obtained as follows:

$$\frac{\partial V_{w0}}{\partial C_p} = \frac{1}{R_p}(\ln\frac{R_P}{C_P} - 1) > 0$$

$$\frac{\partial V_{w0}}{\partial R_p} = -\frac{C_P}{R_P{}^2}(\ln\frac{R_P}{C_P} + 1) < 0$$

End of proof.

Corollary 4.1 shows that, when the cost of making complaints/reports increases, or the rewards provided by new media for making complaints/reports decreases, the probability of the public choosing not to make a complaint/report increases. When the cost of making complaints/reports decreases or the rewards provided by new media for making complaints/reports increases, the probability of the public choosing not to make a complaint/report decreases.

**Corollary 4.2:** The probability of the public choosing to make a complaint/report is negatively correlated to the cost of doing so and positively correlated to the rewards received from new media.

Proof: Based on $V_{w1}$, the first-order partial derivatives of $C_p$ and $R_p$ can be obtained as follows:

$$\frac{\partial V_{w1}}{\partial C_p} = -\frac{1}{R_p}(\ln\frac{C_P}{R_P} - 1) < 0$$

$$\frac{\partial V_{w1}}{\partial R_p} = \frac{C_P}{R_P{}^2}(\ln\frac{R_p}{C_p} + 1) > 0$$

End of proof.

Corollary 4.2 shows that, when the cost of making complaints/reports is greater, the probability of the public choosing to make a complaint/report becomes lower. When the rewards provided by new media increases, the probability of the public choosing to make a complaint/report also increases.

## 5. Stability of strategy combinations

In the replicator dynamic system of the four-party game model, the stability of the strategy combination can be determined using Lyapunov's first stability method. Specifically, if the eigenvalues of the Jacobian matrix are negative, then the equilibrium point is an evolutionarily stable strategy (ESS); if at least one of the eigenvalues is positive, the equilibrium point is an unstable point; and if some of the eigenvalues are zero and the rest are negative, the equilibrium point is at the threshold and stability is uncertain. In this section, the stability of 16 pure-strategy Nash equilibriums is analyzed. Based on the players' replicator dynamics equations,

the Jacobian matrix of the replicator dynamic system is

$$J = \begin{bmatrix} \partial F(x)/\partial x & \partial F(x)/\partial y & \partial F(x)/\partial z & \partial F(x)/\partial w \\ \partial F(y)/\partial x & \partial F(y)/\partial y & \partial F(y)/\partial z & \partial F(y)/\partial w \\ \partial F(z)/\partial x & \partial F(z)/\partial y & \partial F(z)/\partial z & \partial F(z)/\partial w \\ \partial F(w)/\partial x & \partial F(w)/\partial y & \partial F(w)/\partial z & \partial F(w)/\partial w \end{bmatrix}$$

## 5.1 Stability of strategy combinations with new media participation

When the stability strategy of new media is to participate in co-governance, the condition $V_m - C_m > R_p$ is satisfied, and the asymptotic stability of the equilibrium point of the replicator dynamic system is shown in Table 3.

Table 2 shows that, in the context of new media participation, if condition (1) is satisfied, the equilibrium point (0,0,1,1) is the ESS; and if condition (2) is satisfied (the income generated from operating as a monopoly and the cost of operating in compliance with rules and regulations is greater than the sum of the costs of operating as monopoly, the income generated from operating in compliance with rules and regulations and the penalty for monopolistic behavior), then the equilibrium point (1,0,1,1) is the ESS.

## 5.2 Stability of strategy combinations without new media participation

When the stability strategy of new media is not to participate in co-governance, the condition $V_m < C_m + R_p$ is satisfied, and the asymptotic stability of the equilibrium point of the replicator dynamic system is shown in Table 4.

Table 3 shows that, in the context of no new media participation, if condition (3) is satisfied (the cost of the government applying co-governance and supervision is greater than the sum of the cost of applying traditional supervision, information value obtained from the co-governance platform, increase in public trust, and the loss of not being able to identify monopolies), then the equilibrium point (0,0,0,0) is the ESS; and if condition (4) is satisfied (the income generated from operating as a monopoly and the cost of operating in compliance with rules and regulations is greater than the sum of the cost of operating as a monopoly, the income

**Table 3. Asymptotic stability of the equilibrium point of the replicator dynamic system with new media participation.**

| Equilibrium Point | Eigenvalue $\lambda_1,\lambda_2,\lambda_3,\lambda_4$ | Sign | Stability |
|---|---|---|---|
| (0, 0, 1, 0) | $-C_{gh} + C_{gl} + V_g + H_g + \Delta\alpha D_g, R_{el} - R_{eh} + C_{el} - C_{eh} + I_e + \alpha F_e + \alpha N_e, C_m, -C_p + R_p$ | (+,+,+,×) | Unstable |
| (1, 0, 1, 0) | $C_{gh} - C_{gl} - V_g - H_g - \Delta\alpha D_g, R_{el} - R_{eh} + C_{el} - C_{eh} + I_e + \alpha F_e + \Delta\alpha F_e + \alpha N_e + \Delta\alpha N_e, C_m, -C_p + R_p$ | (−,+,+,×) | Unstable |
| (0, 1, 1, 0) | $-C_{gh} + C_{gl}, R_{eh} - R_{el} + C_{eh} - C_{el} - I_e - \alpha F_e - \alpha N_e, C_m, -C_p$ | (−,×,+,−) | Unstable |
| (0, 0, 1, 1) | $-C_{gh} + C_{gl} + V_g + H_g + \Delta\alpha D_g, R_{el} - R_{eh} + C_{el} - C_{eh} + I_e + \alpha F_e + \alpha N_e, C_m - V_m + R_p, C_p - R_p$ | (×,×,−,×) | When ① is satisfied, then ESS |
| (1, 1, 1, 0) | $C_{gh} - C_{gl}, R_{eh} - R_{el} + C_{eh} - C_{el} - I_e - \alpha F_e - \Delta\alpha F_e - \alpha N_e - \Delta\alpha N_e, C_m, -C_p$ | (+,×,+,−) | Unstable |
| (1, 0, 1, 1) | $C_{gh} - C_{gl} - V_g - H_g - \Delta\alpha D_g, R_{el} - R_{eh} + C_{el} - C_{eh} + I_e + \alpha F_e + \Delta\alpha F_e + \alpha N_e + \Delta\alpha N_e - C_m + V_m - R_p, C_p - R_p$ | (−,×,−,×) | When ② is satisfied, then ESS |
| (0, ', 1, 1) | $C_{gh} - C_{gl}, R_{eh} - R_{el} + C_{eh} - C_{el} - I_e - \alpha F_e - z\alpha N_e, C_m, C_P$ | (−,×,+,+) | Unstable |
| (1, 1, 1, 1) | $C_{gh} - C_{gl}, R_{eh} - R_{el} + C_{eh} - C_{el} - I_e - \alpha F_e - \Delta\alpha F_e - \alpha N_e - \Delta\alpha N_e, C_m, C_P$ | (+,×,+,+) | Unstable |

Note: X indicates that the sign cannot be determined; if conditions ① and ② are satisfied, it is a stable point.

Condition ①: $C_m + R_p < V_m$; Condition ②: $-C_m + V_m - R_p < 0$.

**Table 4. Asymptotic stability of the equilibrium point of the replicator dynamic system without new media participation.**

| Equilibrium Point | Eigenvalue $\lambda_1,\lambda_2,\lambda_3,\lambda_4$ | Sign | Stability |
|---|---|---|---|
| (0, 0, 0, 0) | $C_{gl} - C_{gh} + V_g + H_g + \Delta\alpha D_g, R_{el} - R_{eh} + C_{el} - C_{eh} + \alpha F_e, -C_m, -C_p$ | (×,−,−,−) | When ③ is satisfied, then ESS |
| (1, 0, 0, 0) | $C_{gh} - C_{gl} - V_g - H_g - \Delta\alpha D_g, R_{el} - R_{eh} + C_{el} - C_{eh} + \alpha F_e + \Delta\alpha F_e, -C_m, -C_p$ | (−.×,−,−) | When ④ is satisfied, then ESS |
| (0, 1, 0, 0) | $C_{gh} - C_{gl}, R_{eh} - R_{el} + C_{eh} - C_{el} - \alpha F_e, -C_m, -C_P$ | (+,×,−,−) | Unstable |
| (0, 0, 0, 1) | $C_{gl} - C_{gh} + V_g + H_g + \Delta\alpha D_g, R_{el} - R_{eh} + C_{el} - C_{eh} + I_e + \alpha F_e + \alpha N_e, V_m - R_p - C_m, C_P$ | (+,×,−,+) | Unstable |
| (1, 1, 0, 0) | $C_{gh} - C_{gl}, R_{eh} - R_{el} + C_{eh} - C_{el} - \alpha F_e - \Delta\alpha F_e, -C_m, -C_P$ | (+,×,−,−) | Unstable |
| (1, 0, 0, 1) | $-C_{gh} + C_{gl} + V_g + H_g + \Delta\alpha D_g, R_{el} - R_{eh} + C_{el} - C_{eh} + \alpha F_e + \Delta\alpha F_e, -C_m + V_m - R_p, C_P$ | (+,×,−,+) | Unstable |
| (0, 1, 0, 1) | $-C_{gh} + C_{gl}, R_{eh} - R_{el} + C_{eh} - C_{el} - \alpha F_e, -C_m, C_P$ | (−,×,−,+) | Unstable |
| (1, 1, 0, 1) | $C_{gh} - C_{gl}, R_{eh} - R_{el} + C_{eh} - C_{el} - \alpha F_e - \Delta\alpha F_e, -C_m, C_P$ | (+,×,−,+) | Unstable |

Note: × indicates that the sign cannot be determined; if conditions ③ and ④ are satisfied, it is a stable point.

Condition ③: $C_{gh} - C_{gl} > V_g + H_g + \Delta\alpha D_g$; Condition ④: $R_{eh} + C_{eh} > \alpha F_e + \Delta\alpha F_e + C_{el} + R_{el}$.

generated from operating in compliance with rules and regulations and the penalty for monopolistic behavior), then the equilibrium point (1,0,0,0) is the ESS.

## 6. Simulation analysis

To reflect the impact of changes in decision variables on players' decision-making more intuitively, a numerical simulation, using MATLAB 2020b, was conducted to verify the effectiveness of the stability analysis.

We set the cost of the government applying traditional supervision $C_{gl} = 3$, the cost of co-governance and supervision $C_{gh} = 7.5$, the information value obtained from the co-governance platform $V_g = 4.0$, the penalty imposed on monopoly $F_e = 3.0$, the increased public trust $H_g = 1$, and the negative impact of not being able to identify monopolistic behavior $D_g = 1.5$. The income generated by Internet platforms' when operating in compliance with rules and regulations $R_{el} = 3$, the income when operating as a monopoly $R_{eh} = 4$, the cost of operating in compliance with rules and regulations $C_{eh} = 2$, the cost of operating as a monopoly $C_{el} = 1$. The cost of new media participating in co-governance $C_m = 3$, the information value obtained by participating in co-governance $V_m = 2$, the reward given to the public for making complaints/reports $R_p = 1$. The reward given by the new media to the public $R_p = 1$, the cost of making a complaint/report $C_p = 2.5$. Internet platforms' loss of reputation when operating as a monopoly and being exposed by new media $N_e = 2$, whilst the increased reputation generated through positive exposure on new media $I_e = 1$, and the loss of the rights and interests to the public by operating as a monopoly $L_p = 1$. Government's ability to identify monopolistic behavior when applying traditional supervision $\alpha$ was 0.3, while the increase in identification ability when applying co-governance and supervision $\Delta\alpha$ was 0.4.

### 6.1 Influence of the probability of new media participation

We set the probability of new media participating in co-governance $z = \{0.2, 0.5, 0.8\}$. The evolution of the strategies adopted by government, Internet platforms, and the public as well as the results are exhibited in Fig 6.

Without new media participation, the public does not make complaints/reports, and Internet platforms choose to operate as a monopoly. However, as the probability of new media participation in co-governance increases, the strategy selection of the government and Internet platforms becomes unstable. The probability of Internet platforms' strategy selection is constantly adjusted with the probability of the government's strategy selection, and when the

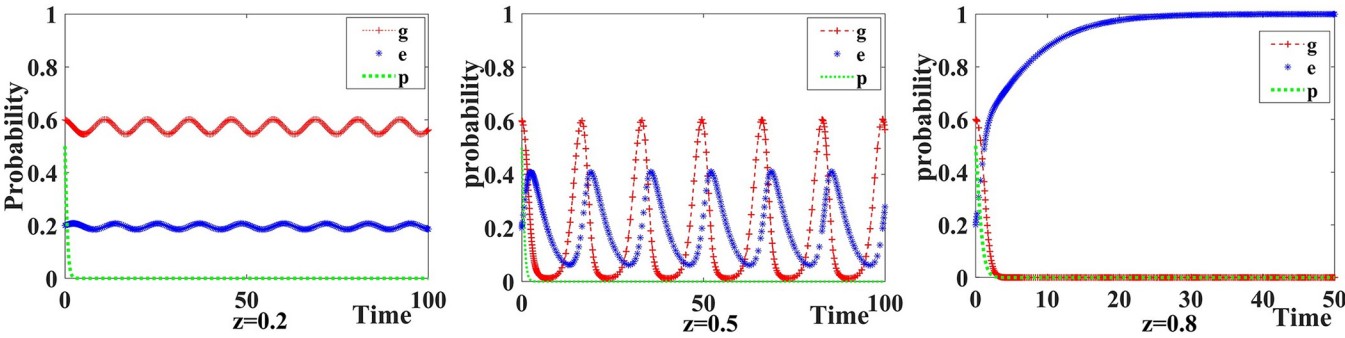

**Fig 6. Influence of the probability of new media participation.**

probability of the government choosing co-governance and supervision is low, the probability of Internet platforms choosing to operate in compliance with rules and regulations decreases. When the probability of new media participation continues to increase, the stable strategy of Internet platforms shifts to operate in compliance with the rules and regulations. At this time, the public does not make complaints/reports, and the government does not need to apply co-governance and supervision. In that sense, the participation of new media in co-governance plays an effective supervisory role in inhibiting monopolistic operations of Internet platforms.

## 6.2 Influence of new media participation on reputation value

We set the increase and loss in Internet platforms' reputation value caused by participation of new media $I_e = \{0.0, 2.0, 4.0\}$ and $N_e = \{0.0, 2.0, 4.0\}$ respectively. The evolution of the strategies adopted by government, Internet platforms, New Media, and the Public as well as the results are exhibited in Fig 7.

When the influence of new media participation on Internet platforms is lacking, the probability of companies choosing to operate in compliance with the rules and regulations fluctuates with the probability of the government choosing to apply co-governance and supervision. Thus, the strategies of the government and Internet platforms fluctuate constantly, and no stable strategies are reached. At this stage, new media chooses not to participate in co-governance and the public chooses not to make complaints/reports. When new media participation has limited effect on the changes in Internet platforms' reputation value, the government tends to prefer co-governance and supervision to control monopolistic behaviors. At this stage, the stable strategy of the government is co-governance and supervision and that of Internet platform companies is operating in compliance with rules and regulations.

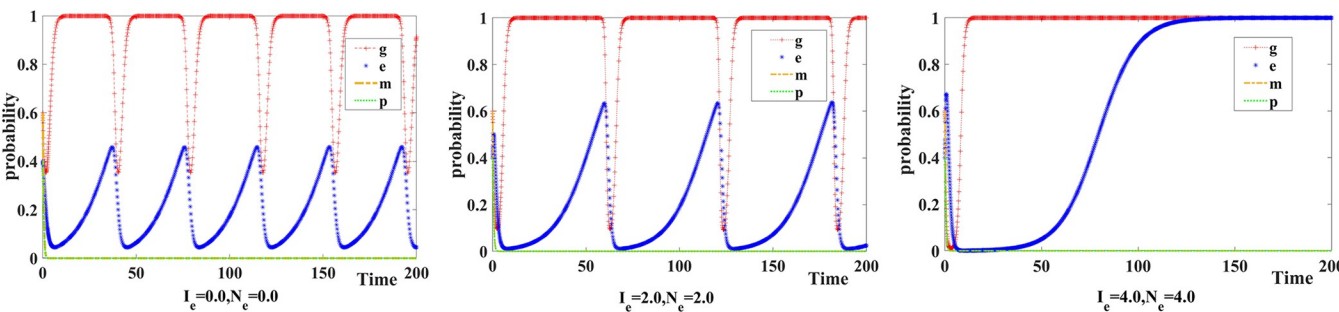

**Fig 7. Influence of new media participation on reputation value.**

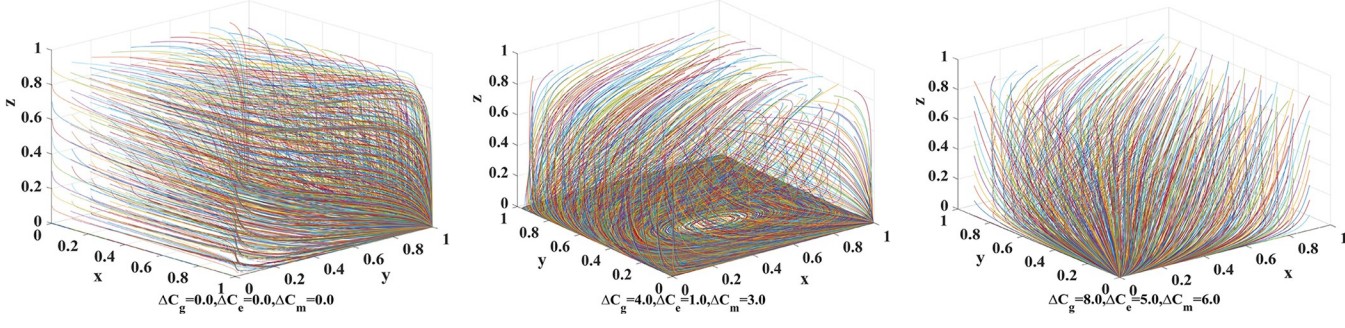

**Fig 8. Influence of additional costs.**

## 6.3 Influence of additional costs

We set the probability of the public choosing to make a complaint/report $r = 0.5$, the additional cost of the government choosing to apply co-governance and supervision $\Delta C_g = \{0.0, 4.0, 8.0\}$, the additional cost of Internet platforms choosing to operate in compliance with rules and regulations $\Delta C_e = \{0.0, 1.0, 5.0\}$, and the additional cost of new media choosing to participate in co-governance $\Delta C_m = \{0.0, 3.0, 6.0\}$. Fig 8 shows the evolution of the strategies adopted by government, Internet platforms, and new media as well as the results.

When the additional costs of applying co-governance and supervision (government), operating in compliance with rules and regulations (Internet platforms), and participating in co-governance (new media) are high, there is a stable equilibrium point (0,0,0) in the replicator dynamic system. At this stage, the government chooses the traditional supervision system, Internet platforms choose to operate as a monopoly, and new media chooses not to participate in co-governance. As the players' additional costs reduce, the stable equilibrium point of the system becomes (1,1,0), where the government chooses co-governance and supervision, Internet platforms choose to operate in compliance with rules and regulations, and new media chooses not to participate in co-governance.

## 7. Conclusions and suggestions

The monopolistic operations of Internet platform companies seriously affect fair competition and the development of the market economy, leaving negative impacts on market order and social stability and development. Constructing and reinforcing a co-governance and supervision system that engages social forces is conducive to controlling and inhibiting monopolies in the market. Compliance operations and sustainable and healthy development of Internet platforms require joint supervision from the government, the public and new media. This study investigated the anti-monopoly co-governance and supervision system in the context of new media participation and constructed a four-party evolutionary game model to explore the stable equilibrium points of each player's strategy selection and the stability of strategy combinations in the replicator dynamic system. Then, simulation analysis was performed using MATLAB 2021b to examine the impact of changes in decision variables on the evolution of players' strategy selection. Based on the findings, we proposed the following suggestions:

1. Actively adopt a co-governance and supervision strategy: The government should utilize the Internet and big data technology to form a supervision system that encourages the companies and the public to jointly participate in anti-monopoly campaigns, facilitate timely supervision, and promote the construction of the informatization of co-governance, to maintain the orderly development of the market.

2. More effective reduce participation costs: The cost for Internet platforms to participate in co-governance should be reduced to stimulate their engagement and promote operations in compliance with rules and regulations. The costs of applying co-governance and supervision should be reduced to boost enthusiasm of the government departments in anti-monopoly governance, and to improve their regulatory capability, thereby ensuring a healthy market environment for orderly competition and protecting the interests of the public.

3. Actively encourage participation of the new media: This study found that, when the probability of new media choice to participate in co-governance is high, Internet platforms are more likely to choose to operate in compliance with rules and regulations; when new media chooses not to participate in co-governance, Internet platforms' strategy selection becomes unstable and carries a higher likelihood of choosing not to participate in the co-governance and supervision system. The greater the impact of new media participation, the greater the impact of new media on the reputation value of Internet platforms. In this situation, the stable strategy for Internet platforms shifts to operating in compliance with rules and regulations, while the government's enthusiasm for applying co-governance and supervision increases.

4. Apply penalties to effectively increase the enthusiasm of the government, Internet platforms, new media, and the public to participate in co-governance and supervision. Severe penalties were found to be effective at stimulating Internet platforms' engagement in anti-monopoly strategies, strengthening government supervision on their economic behaviors, and urging platforms to actively participate in anti-monopoly supervision.

In this study, we constructed an evolutionary game model involving the participation of the government, Internet platforms, new media, and the public, to explore the impact of new media on other players' strategy selection. However, the model was single stage and constructed under bounded rationality with complete information, without considering the sequence of the players' behaviors. A multi-player, multi-stage, repeated, dynamic model under conditions of incomplete information is planned for subsequent analysis.

## Supporting information

**S1 File.**
(DOCX)

## Acknowledgments

The authors are grateful to the referees for their valuable comments and their helps on how to improve the quality of our paper.

## Author Contributions

**Conceptualization:** Yadong Song, Tongshui Xia.

**Data curation:** Yadong Song.

**Formal analysis:** Yadong Song.

**Methodology:** Yadong Song.

**Software:** Yadong Song.

**Supervision:** Tongshui Xia.

**Validation:** Yadong Song.

**Visualization:** Tongshui Xia.

**Writing – original draft:** Yadong Song.

**Writing – review & editing:** Yadong Song, Tongshui Xia.

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
