## [Decision Letter · Decision Letter 0]

18 Dec 2023

PONE-D-23-39623Players’ Strategy Selection in Co-Governance and Supervision of Internet Platforms’ Monopolistic Behaviors: A Study on New Media ParticipationPLOS ONE

Dear Dr. Song,

Thank you for submitting your manuscript to PLOS ONE. After careful consideration, we feel that it has merit but does not fully meet PLOS ONE’s publication criteria as it currently stands. Therefore, we invite you to submit a revised version of the manuscript that addresses the points raised during the review process.

We look forward to receiving your revised manuscript.

Kind regards,

Dr. Jiachao Peng

Academic Editor

PLOS ONE

[The authors are grateful to the referees for their valuable comments and their helps on 

how to improve the quality of our paper. This work was supported by the National 

Social Science Fund of China under grant Nos.20BGL272 and 21ZDA024, the Nature 

Science Foundation of Shandong Province in China under grant No.ZR2019MG017, 

and the University Youth Science and Technology Innovation Team Project of 

Shandong Province in China under grant No. 2021RW010.]

 [The author(s) received no specific funding for this work.]

5. PLOS requires an ORCID iD for the corresponding author in Editorial Manager on papers submitted after December 6th, 2016. Please ensure that you have an ORCID iD and that it is validated in Editorial Manager. To do this, go to ‘Update my Information’ (in the upper left-hand corner of the main menu), and click on the Fetch/Validate link next to the ORCID field. This will take you to the ORCID site and allow you to create a new iD or authenticate a pre-existing iD in Editorial Manager. Please see the following video for instructions on linking an ORCID iD to your Editorial Manager account: " ext-link-type="uri" xlink:type="simple">https://www.youtube.com/watch?v=_xcclfuvtxQ".

Reviewers' comments:

Reviewer's Responses to Questions

**Comments to the Author**

1. Is the manuscript technically sound, and do the data support the conclusions?

Reviewer #1: Yes

Reviewer #2: Yes

Reviewer #3: Yes

2. Has the statistical analysis been performed appropriately and rigorously? 

Reviewer #1: Yes

Reviewer #2: Yes

Reviewer #3: Yes

3. Have the authors made all data underlying the findings in their manuscript fully available?

Reviewer #1: Yes

Reviewer #2: Yes

Reviewer #3: Yes

4. Is the manuscript presented in an intelligible fashion and written in standard English?

Reviewer #1: Yes

Reviewer #2: Yes

Reviewer #3: Yes

5. Review Comments to the Author

Reviewer #1: This article is a hot issue of concern to everyone now in the current climate. Internet platform enterprise business events are related to the development of market economy, public interest and social fairness and other issues. Responding to the antitrust incidents of Internet platform enterprises, this article introduces multi-party subjects to participate in the governance and supervision based on the establishment of a perfect antitrust supervision system by government departments. It also constructs an evolutionary game model involving government departments, Internet platform enterprises, new media and the public, solves the stable equilibrium point of strategy selection of each game subject, analyzes the stability of strategy combinations through Lyapunov's first law, and carries out a simulation analysis using Matlab 2021b to validate the influence of each decision-making variable on the strategy selection of different subjects. Overall, the article has strong research value and realistic research significance, the research process and logical framework is rigorous and reasonable, and the line is relatively standardized and readable.

However, the current manuscript still needs to answer the following questions and make corresponding improvements:

1. The cases in the introduction need to indicate the source, increase the authenticity and rigor of the event

2. The revision of legal documents should be added as much as possible to indicate the authenticity of the event.

3. Before modeling, please explain the relevance of using evolutionary game to solve the antitrust problem of Internet platform enterprises.

4. Equation (5) and Equation (12) pay attention to the range of values at the ends of [0, 1], which need to be modified, do not repeat the values.

5. Whether the results of the model proof and the results of the simulation analysis can be further combined to make the explanation clearer.

6. Whether the conclusion part of the article can be combined with the reality of antitrust behavior to make more targeted suggestions

In summary, it is recommended that the authors revise the article before sending it for review.

Reviewer #2: This manuscript constructed an evolutionary game model involving the government, Internet platforms, new media, and the public to explore the stable equilibrium point of players’ strategy selections. The stability of the strategy combinations was tested using Lyapunov’s first stability method, and MATLAB 2021b was used to conduct simulation analysis of the impact of each decision variable on players’ strategy selection. This manuscript aims to construct an evolutionary game theory model involving the above four players within the new media environment context to explore the stable equilibrium point of their strategy selection, analyze the stability of their strategy combination, and the impact of changes in decision variables on strategy selection.

The topic of this manuscript has both theoretical and practical significance. The established model is scientific, Data analysis is feasible. The discussion and conclusion have guiding function.

There are three suggestions that the author needs to continue revising:

First, Please the author to cite the latest literature in this field from the past three years 2021-2023.

Second, The manuscript is 35 pages long, please compress it appropriately by the author.

Finally, Please the author to improve the clarity of the Figures to achieve above 300*300 DPI.

Reviewer #3: This article studies the antitrust regulation strategy of Internet platform enterprises under the participation of new media. The antitrust regulation of Internet platform enterprises studied in this article is a hot issue of concern nowadays, which has strong practical significance and research value. This article constructs a four-party evolutionary game model to study the problem from four related subjects: government regulators, Internet platform enterprises, new media and the public, and introduces variables related to the efficiency of co-regulation. The stability of each participant's strategy choice is analyzed, and Matlab2021b is used to conduct numerical simulation to explore the influence of parameter changes on the game behavior of the four participants. Overall, the derivation process and logical design of the article are more complete, the derivation process and the conclusion have a logical echo relationship, and the line is more standardized, the method is properly selected, and the problem awareness is stronger.

However, the current manuscript still needs to answer the following questions and make corresponding improvements:

1. The relevant literature review of the previous part of the lack of important literature with game theory research as a theoretical support, should be added in the literature review of game theory research related literature.

2. In the chapter of model assumption and construction, please explain why the method of evolutionary game is used to analyze and solve this problem.

3. Whether the data selection in the simulation analysis can be realized by using the real data now?

4. It is suggested to read through the article again to improve and enhance the readability of the article.

To summarize, it is recommended that the authors revise the article before sending it for review.

6. PLOS authors have the option to publish the peer review history of their article (what does this mean?). If published, this will include your full peer review and any attached files.

Reviewer #1: No

Reviewer #2: No

Reviewer #3: No

---

## [Author Response · Author response to Decision Letter 0]

24 Jan 2024

Responses to Editor and Reviewers

Dear Editor,

Thank you very much for your critical reading and valuable comments about our paper submitted to PLOS ONE (Manuscript ID: PONE-D-23-39623). The manuscript’s title is “Players’ Strategy Selection in Co-Governance and Supervision of Internet Platforms’ Monopolistic Behaviors: A Study on New Media Participation”.

Those comments were very helpful for providing direction for our further studies. We have tried our best to revise our manuscript according to the comments. Attached, please find the PONE-D-23-39623 Revised Manuscript, which we would like to resubmit for your kind consideration. The following is a detailed explanation how we have complied with the reviewers’ suggestions.

Responds to the reviewers’ comments:

Reviewer 1

Comment #1: 

(1) The cases in the introduction need to indicate the source, increase the authenticity and rigor of the event.

Response: Thank you very much for pointing out the problem. We have indicated the source of the cases in the introduction to increase the authenticity and rigor of the event. such as, In response to antitrust incidents involving Internet platforms, the United States has tightened previously relaxed policies on the platform economy (www.justice.com).

China has also focused on improving regulatory systems and mechanisms, innovating supervision concepts and methods, and developing highly knowledgeable and professional teams to enhance its anti-monopoly regulatory capabilities(www.gov.cn).

We have marked the modification in red in PONE-D-23-39623 Revised Manuscript.

Comment #2: 

(2) The revision of legal documents should be added as much as possible to indicate the authenticity of the event.

Response: Thank you very much for pointing out the problem. We have Add the date of enactment of the law. Such as "The Fair Online Platform Intermediary Transactions Act, drafted by the Korea Fair Trade Commission, requires Internet platforms to provide a written contract that specifies the terms directly related to sellers’ interests during intermediary transactions in March 2021. ","the Competition Commission, which conducts comprehensive anti-monopoly investigations on platform companies; and the Competition Appeal Court) in November 18, 2021. " We have marked the modification in red in PONE-D-23-39623 Revised Manuscript.

Comment #3: 

(3) Before modeling, please explain the relevance of using evolutionary game to solve the antitrust problem of Internet platform enterprises.

Response: Thank you very much for the valuable suggestions. In order to increase the logic of the paper, at the beginning of Chapter 3, we explained the reasons for choosing this research method. "This paper chooses evolutionary game as the research method because it can explain the dynamic process of the evolution of each stakeholder's strategic choice and explain why this state has been reached and how to reach. Therefore, considering a conceptual model of the anti-monopoly co-governance and supervision system, involving the government, Internet platforms, new media, and the public, is presented in Figure 1." We also marked modification in red in PONE-D-23-39623 Revised Manuscript.

Comment #4: 

(4) Equation (5) and Equation (12) pay attention to the range of values at the ends of [0, 1], which need to be modified, do not repeat the values.

Response: Thank you very much for your careful review work. We modify the value range at the end of [0,1] from [0,1] to (0,1). After the modification, the value range does not contain repeated values. We have marked the modification in red in PONE-D-23-39623 Revised Manuscript.

Comment #5:

(5) Whether the results of the model proof and the results of the simulation analysis can be further combined to make the explanation clearer.

Response: Thank you very much for the valuable suggestions. We have added analysis in the simulation analysis section. We have marked the modification in red in PONE-D-23-39623 Revised Manuscript.

Comment #6:

(6) Whether the conclusion part of the article can be combined with the reality of antitrust behavior to make more targeted suggestions.

Response: Thank you very much for the valuable suggestions. We have marked the modification in red in PONE-D-23-39623 Revised Manuscript.

Reviewer 2

Comment #1: 

(1) Please the author to cite the latest literature in this field from the past three years 2021-2023.

Response: Thank you very much for the valuable suggestions. We have cited the latest literature in this field from the past three years 2021-2023.We have marked the modification in red in PONE-D-23-39623 Revised Manuscript.

Comment #2:

(2) The manuscript is 35 pages long, please compress it appropriately by the author.

Response: Thank you very much for the valuable suggestions. We have read through, revised, condensed and compressed the article many times. After re-reading and summarizing, the article has been compressed.

Comment #3:

(3) Please the author to improve the clarity of the Figures to achieve above 300*300 DPI.

Response: Thank you very much for the valuable suggestions. We have processed the picture according to your request and we have improved the clarity of the Figures to achieve above 300*300 DPI.

Reviewer 3

Comment #1:

(1) The relevant literature review of the previous part of the lack of important literature with game theory research as a theoretical support, should be added in the literature review of game theory research related literature.

Response: Thank you very much for the valuable suggestions. We have added in the literature review of game theory research related literature. Such as“Daniel Friedman. On economic applications of evolutionary game theory[J]Journal of Evolutionary Economics,1998(1)”“Alexander JMK. Evolutionary game theory[J].Elements in Decision Theory and Philosophy, 2023.”Owen G. Game theory[M]. Emerald Group Publishing, 2013.“We have marked the modification in red in PONE-D-23-39623 Revised Manuscript.

Comment #2:

(2) In the chapter of model assumption and construction, please explain why the method of evolutionary game is used to analyze and solve this problem.

Response: Thank you very much for the valuable suggestions. In order to increase the logic of the paper, at the beginning of Chapter 3, we explained the reasons for choosing this research method. "This paper chooses evolutionary game as the research method because it can explain the dynamic process of the evolution of each stakeholder's strategic choice and explain why this state has been reached and how to reach. Therefore, considering a conceptual model of the anti-monopoly co-governance and supervision system, involving the government, Internet platforms, new media, and the public, is presented in Figure 1." We also marked modification in red in PONE-D-23-39623 Revised Manuscript.

Comment #3:

(3) Whether the data selection in the simulation analysis can be realized by using the real data now?

Response: Thank you very much for your question. Here we use numerical simulation, combining the real situation and the researches of other scholars to set the values, and achieve the purpose of problem research through numerical calculation and image display. At present, it is impossible to obtain actual data ideally, which is also a problem that we need to work hard to overcome in the future. 

Comment #4:

(4) It is suggested to read through the article again to improve and enhance the readability of the article.

Response: Thank you very much for the valuable suggestions. I have read through the article again to improve and enhance the readability of the article. 

We have tried our best to improve the manuscript and made some changes according to the reviewers’ comments. We earnestly appreciate reviewers’ professional work and hope that the corrections will make our manuscript suitable for publication in PLOS ONE. We are looking forward to receiving comments from reviewers in the future.

Once again, thank you very much for your valuable comments and suggestions.

Best wishes.

Sincerely yours,

Tongshui Xia

13, Jan, 2024

---

## [Decision Letter · Decision Letter 1]

5 Feb 2024

Players’Strategy Selection in Co-Governance and Supervision of Internet Platforms’ Monopolistic Behaviors: A Study on New Media Participation

PONE-D-23-39623R1

Dear Dr. Xia,

We’re pleased to inform you that your manuscript has been judged scientifically suitable for publication and will be formally accepted for publication once it meets all outstanding technical requirements.

Kind regards,

Dr. Jiachao Peng

Academic Editor

PLOS ONE

Additional Editor Comments (optional):

Reviewers' comments:

Reviewer's Responses to Questions

**Comments to the Author**

1. If the authors have adequately addressed your comments raised in a previous round of review and you feel that this manuscript is now acceptable for publication, you may indicate that here to bypass the “Comments to the Author” section, enter your conflict of interest statement in the “Confidential to Editor” section, and submit your "Accept" recommendation.

Reviewer #1: All comments have been addressed

Reviewer #2: All comments have been addressed

Reviewer #3: All comments have been addressed

2. Is the manuscript technically sound, and do the data support the conclusions?

Reviewer #1: Yes

Reviewer #2: Yes

Reviewer #3: Yes

3. Has the statistical analysis been performed appropriately and rigorously? 

Reviewer #1: Yes

Reviewer #2: Yes

Reviewer #3: Yes

4. Have the authors made all data underlying the findings in their manuscript fully available?

Reviewer #1: Yes

Reviewer #2: Yes

Reviewer #3: Yes

5. Is the manuscript presented in an intelligible fashion and written in standard English?

Reviewer #1: Yes

Reviewer #2: Yes

Reviewer #3: Yes

6. Review Comments to the Author

Reviewer #1: In the revised version, the author has addressed each of the comments in their response. Therefore, I do think that this article could be accepted and published in PLOS ONE.

Reviewer #2: Firstly, The authors have adequately addressed my comments.

Secondly, after revising the manuscript item by item, the level and quality of the manuscript have greatly improved, meeting the requirements for journal publication.

Finally, I suggest ACCEPT the manuscript for journal publication.

Reviewer #3: I am very glad to receive the “Responses for the Comments” soon. All comments have been addressed. Therefore, I do think that this article could be accepted and published in PLOS ONE.

7. PLOS authors have the option to publish the peer review history of their article (what does this mean?). If published, this will include your full peer review and any attached files.

Reviewer #1: No

Reviewer #2: No

Reviewer #3: No

---

## [Editor Report · Acceptance letter]

26 Feb 2024

PONE-D-23-39623R1 

PLOS ONE

Dear Dr. Xia, 

I'm pleased to inform you that your manuscript has been deemed suitable for publication in PLOS ONE. Congratulations! Your manuscript is now being handed over to our production team.

Kind regards, 

on behalf of

Dr. Jiachao Peng 

Academic Editor

PLOS ONE